# Meta-Analysis of Human and Mouse Biliary Epithelial Cell Gene Profiles

**DOI:** 10.3390/cells8101117

**Published:** 2019-09-20

**Authors:** Stefaan Verhulst, Tania Roskams, Pau Sancho-Bru, Leo A. van Grunsven

**Affiliations:** 1Liver Cell Biology Research Group, Department of Basic Biomedical Sciences, Vrije Universiteit Brussel, 1090 Brussels, Belgium; Stefaan.verhulst@vub.be; 2Department of Imaging and Pathology, Translational Cell and Tissue Research, KU Leuven and University Hospitals, 3000 Leuven, Belgium; tania.roskams@kuleuven.be; 3Institut d’Investigacions Biomediques August Pi i Sunyer (IDIBAPS), 08001 Barcelona, Spain; PSANCHO@clinic.cat

**Keywords:** BEC, transcriptome, scRNAseq, gene signature

## Abstract

Background: Chronic liver diseases are frequently accompanied with activation of biliary epithelial cells (BECs) that can differentiate into hepatocytes and cholangiocytes, providing an endogenous back-up system. Functional studies on BECs often rely on isolations of an BEC cell population from healthy and/or injured livers. However, a consensus on the characterization of these cells has not yet been reached. The aim of this study was to compare the publicly available transcriptome profiles of human and mouse BECs and to establish gene signatures that can identify quiescent and activated human and mouse BECs. Methods: We used publicly available transcriptome data sets of human and mouse BECs, compared their profiles and analyzed co-expressed genes and pathways. By merging both human and mouse BEC-enriched genes, we obtained a quiescent and activation gene signature and tested them on BEC-like cells and different liver diseases using gene set enrichment analysis. In addition, we identified several genes from both gene signatures to identify BECs in a scRNA sequencing data set. Results: Comparison of mouse BEC transcriptome data sets showed that the isolation method and array platform strongly influences their general profile, still most populations are highly enriched in most genes currently associated with BECs. Pathway analysis on human and mouse BECs revealed the KRAS signaling as a new potential pathway in BEC activation. We established a quiescent and activated BEC gene signature that can be used to identify BEC-like cells and detect BEC enrichment in alcoholic hepatitis, non-alcoholic steatohepatitis (NASH) and peribiliary sclerotic livers. Finally, we identified a gene set that can distinguish BECs from other liver cells in mouse and human scRNAseq data. Conclusions: Through a meta-analysis of human and mouse BEC gene profiles we identified new potential pathways in BEC activation and created unique gene signatures for quiescent and activated BECs. These signatures and pathways will help in the further characterization of this progenitor cell type in mouse and human liver development and disease.

## 1. Introduction

Chronic liver diseases (CLD) can lead to hepatic dysfunction with organ failure. Early studies in humans showed that in conditions of submassive necrosis, reactive ductules and intermediate hepatocyte-like cells originate from the activation and differentiation of putative progenitor cells [1,2]. In such conditions, adult biliary epithelial cells (BECs) are believed to activate and differentiate, thereby providing an endogenous back-up system for replenishing hepatocytes and cholangiocytes when the regenerative capabilities of these cells are impaired [3,4,5]. When the biliary regeneration is compromised, hepatocytes can also provide a backup mechanism by transdifferentiating into BECs [6,7].

Due to their capacity for long-term expansion, chromosomal stability and their differentiation potential towards hepatocytes, such BECs could provide an exciting alternative over primary hepatocytes for toxicological studies and use in regenerative medicine [8]. Still, it is unclear whether BECs significantly contribute to liver regeneration. Studies based on in vivo lineage tracing in mice [9,10] initially revealed that only a limited percentage of BEC-derived hepatocytes could be detected during liver regeneration, suggesting a low contribution of BECs. Later studies show that, under very specific liver injury conditions in the mouse, BECs can significantly contribute to the regeneration of the liver [11,12]. In contrast, other studies have shown that inhibition of BEC regeneration impairs liver recovery and decreases survival [13,14].

The functionality of BECs can be studied by analyzing their transcriptome after isolation from healthy and injured mouse livers due to dietary supplements that result in chronic liver injury; the DDC (3,5-diethoxycarbonyl-1,4-dihydrocollidine) [15] and CDE (choline-deficient, ethionine-supplemented) [16] diets are the most frequently used models to establish an activation of the BECs (also known as a ductular reaction). The DDC diet is metabolized by hepatocytes into toxic protoporphyrins that are secreted into the bile ducts leading to cholangitis. This results in BEC expansion around the portal vein and differentiation into cholangiocytes [17]. A CDE diet results in hepatic damage with the formation of a ductular reaction going from the portal vein to the parenchyma and BEC differentiation towards hepatocytes [16]. BECs that can activate are sometimes also referred to as liver progenitor cells. In this manuscript we will refrain from using liver progenitor cells as a term and instead will refer to quiescent and activated BECs, depending on the state of the livers from which the BECs were isolated (healthy vs. diseased). For the isolation of BECs, surface markers [18,19], functional assays [20,21] or BEC reporter mice [10,22,23] are popular methods. While many studies have generated transcriptome profiles of BECs, isolated using different approaches from different mouse injury models (Table 1), so far, no comparative study has been conducted to compare gene signatures of human and mouse BECs. It is not unlikely that the use of different BEC isolation techniques has led to the isolation of subsets of BECs, which can lead to contradicting results if one considers them as the same.

An important hurdle to tackle in such a comparative study is that none of the publicly available transcriptome data sets of primary BECs use the same BEC marker or reporter gene for the isolation these cells. Indeed, Hnf1β [10], Mic1 1c3 [18] (further referred as Mic1c3), Foxl1 [22], Lgr5 [23] and Epcam [24] are all used to isolate BECs, but until recently there were no independent studies that confirmed any of these gene expression profiles. The Hnf1β^+^-, Foxl1^+^- and Lgr5^+^- BECs were isolated based on lineage tracing whereas Mic1c3^+^- and Epcam^+^-BECs were isolated based on their expression on the surface of these cells. One has to note that Hnf1β, Mic1c3 and Epcam are expressed in cholangiocytes as well as in quiescent BECs, while Foxl1 and Lgr5 are mainly expressed in activated BECs [10,18,22,23,24]. In addition, Foxl1 and Mic1c3 have been used to isolate activated BECs from DDC-injured livers, while Hnf1β positive cells were isolated from both CDE and DDC injured livers (Table 1). Other toxin-based models to study BEC biology have rarely been used to isolated BECs from, with exception of Lgr5^+^ cells isolated after a single CCl_4_ injection [23].

In a previous study, we reported on the first RNA sequencing-based transcriptome profiles of BECs isolated from alcoholic steatohepatitis patients through EpCAM- or TROP2-based FACS sorting (respectively epithelial cell adhesion molecule (TACSTD1) and trophoblast antigen 2 (TACSTD2) [25]. TROP-2 is a relatively new epithelial marker and is specifically expressed by activated progenitor cells in mouse models of liver disease [19]. Epcam is a well-established BEC marker which identifies cholangiocytes as well as BECs in mice and humans [26].

In this study, we performed a meta-analysis on gene expression data sets of both human and mouse BECs and created unique signatures for quiescent and activated BECs. Gene set enrichment analysis using these BEC signatures revealed an enrichment in livers of alcoholic steatohepatitis (ASH), non-alcoholic steatohepatitis (NASH) and primary sclerosing cholangitis (PSC) suggesting that these diseases are accompanied by a strong BEC activation. Finally, a selection of our BEC gene signatures can be used to identify quiescent and activated BECs in single cell RNA sequencing (scRNA seq) data sets.

## 2. Materials and Methods

### 2.1. Source of Gene Expression Data 

We searched for publicly available transcriptomic data sets of BECs (see Table 1). We included only microarray data for mouse BECs, with at least two biological repeats, and using one of the three most widely used microarray platforms (Agilent 014868, Mouse 430_2 and Mogene 2.0st) so as to simplify the experiment. We avoided including more microarray platforms to avoid loss of genes due to mismatched annotation between the different platforms.

All microarray and RNA seq data used in this study are publicly available and described in Table 1 and Table 2. Raw microarray files were downloaded from NCBI (https://www.ncbi.nlm.nih.gov/geo) and imported into RStudio (https://www.rstudio.com). For human BEC RNA sequencing data, normalized count files were used from Ceulemans et al. (Table 1) [25]. BEC scRNA seq data was used from Pepe-Mooney et al., 2019 (GSE125688) and Azarina et al., 2019 (GSE124395) [27,28]. Appendix A shows a list of samples that refer to quiescent or activated BECs.

### 2.2. Microarray Data Preparation

Microarray date sets were imported separately in RStudio and normalized using Robust Multiarray Averaging using R packages “affy” [31] and “limma” [32] and duplicated gene symbols were removed. Next, all datasets were pooled together based on their gene symbol and normalized a second time to decrease batch effects using Cycle Loess algorithm. Correlation analysis is performed on merged data with tSNE plot (R package “Rtsne”) and Pearson correlation heatmap in RStudio.

### 2.3. Generation of BEC Gene Signatures 

First, mouse BEC transcriptome data was compared to healthy liver transcriptome data and genes were selected by a fold change larger than 8 and corrected *p* value lower than 0.05 using a Benjamini–Hochberg test. Next, genes were selected by comparing BEC transcriptomes to multiple cell types with criteria used in Friedmann et al., (fold change and *p* value) [33]. BEC signatures were obtained by merging both gene sets with those of human BEC signatures from Ceulemans et al. [25] using Venn diagrams (R package “VennDiagram”).

### 2.4. Gene Set Enrichment Analysis 

Gene set enrichment analysis (GSEA) analysis was performed on normalized intensity values (microarray) or counts (RNA seq, transcripts per million) by comparing healthy livers (mouse data) or injured livers (human data) versus BEC transcriptomes. All Hallmark pathways were analyzed, and false discovery rate (FDR) scores were imported into RStudio to visualize, using heatmaps (R package “caret”). Significantly enriched pathways were based on positive NES score and FDR < 0.25 in at least one population. GSEA analysis to test BEC signatures were visualized using R package “circlize” by displaying -log(FDR) with a maximum -log(FDR) equal to 4 (FDR < 0.0001) for optimal visualization purposes. The direction of arrows represents enrichment of a signature towards cell types or liver tissues. Size of the arrow represents -log(FDR).

### 2.5. Gene Ontology Analysis 

GO analysis from quiescent and activation BEC gene signature was obtained using R package “clusterProfiles” and human database from R package “AnnotationHub”. All biological processes were analyzed with p cutoff of 0.05. GO were visualized using the “dotplot” function in clusterProfiles.

### 2.6. Single Cell Signature Explorer 

ScRNA seq data of BECs and Hepatocytes were downloaded from GEO database (GSE125688) and imported into RStudio. TSNE plots were created using “Seurat” packages [34]. Gene signature scores were calculated and visualized using “Single-Cell Signature Explorer” (https://sites.google.com/site/fredsoftwares/products/single-cell-signature-explorer). Briefly, gene signature scores are computed by Single-Cell Signature Score in linux. TSNE1 and tSNE2 values created within Seurat are merged together with signature score for each cell using Single-Cell Signature Merger and imported in RStudio. Single-Cell Signature Viewer, a shiny app (https://shiny.rstudio.com), was used to visualize signature scores on tSNE plots with adjustable scale bar.

## 3. Results

### 3.1. BEC Transcriptome Profiles Are Highly Affected by the Microarray Platform and Markers Used for Isolation

To establish comparable mouse BEC gene expression data sets, we first normalized each set separately and then pooled all sets together and eventually normalized the complete pooled set to minimize batch effects (Figure 1A). To be able to merge all of the microarrays, we first had to exclude some genes, for several reasons. Each microarray platform detects more than 20,000 genes by using probes that can bind to specific genes or even multiple genes. In our analysis, we first discarded probes that bind on multiple genes and afterwards discarded other genes that are not detected by all microarray platforms. We also noted that we lost several genes because multiple microarray platforms annotate some genes with different gene symbols. Finally, by pooling all microarrays, we obtained a dataset that contained 12,873 genes. 

We first compared the expression profiles of freshly isolated mouse BECs [10,18,22,23,24,29] with four BEC cell lines [30] and healthy liver transcriptome data [30]. Multidimensional reduction analysis, presented in a t-distributed stochastic neighbor embedding (t-SNE) plot, showed that these three groups were clustered separately from each other (Figure 1B). We observed that within the primary BEC group, quiescent and activated BECs isolated using the same approach, could not be distinguished from each other (triangles cluster with circles of the same color; Figure 1B). Furthermore, it was striking to see that Hnf1b^+^ and EpCam^+^ BEC profiles were separated from the other primary BECs in this tSNE plot. This was confirmed by Pearson correlation analyses (Figure 1C); Epcam^+^ BECs cluster together with BEC cell lines and healthy livers while Hnf1β^+^ cells are again separated from these. A closer look at the platform taught us that Epcam^+^ BECs and the BEC cell lines were analyzed on the same Agilent platform, while Hnf1β^+^ cells were the only BECs analyzed with the Affimetrix (mogene 2.0 array). Together, this analysis confirms that the isolation method (or experimenter) and array platforms used are very strong confounders when different data sets need to be compared [35,36].

### 3.2. Meta-Analysis Confirms the Validity of Most BEC-Specific Genes

Despite these platform- and isolation-specific effects, we wanted to know whether the primary mouse and human BEC populations are indeed enriched in commonly used BEC-specific genes. As expected, virtually all commonly used BEC markers (highlighted by Rodrigo-Torres et al. [10]) are highly enriched in all the different mouse and human BEC populations (Figure 1D). The only exception is the Lgr5^+^ BEC population isolated from CCl_4_-treated mice, which suggests that Lgr5^+^ cells isolated from these livers are not similar to any of the other BECs isolated through other methods from other liver injury models. 

The role of BECs in healthy and diseased livers has been intensively studied by lineage tracing experiments in mice using BEC-specific reporter mice, such as Krt19, Sox9, Foxl1, Lgr5, Spp1 (Opn) or Hnf1β [9,12,14,22,23,37]. We therefore also analyzed their gene expression in these human and mouse BEC populations. KRT19, SOX9, SPP1 and HNF1β are greatly enriched in almost all mouse and human BEC populations confirming that these genes are indeed good candidates to be used as BEC-specific driver genes. Lgr5^+^ BECs do not show enriched expression of these genes and even under represents SOX9, suggesting again that Lgr5 positivity does not identify a traditional BEC population (Figure 2). This notion is strengthened by Lgr5 gene expression, which is only enriched in the Lgr5^+^ BEC population and not in other BEC populations. In contrast, the mouse Epcam^+^ population also expresses poorly SOX9 and HNF1β but the Epcam gene is highly expressed in all human and mouse BEC populations. A possible explanation could be the isolation procedure and analysis platform used, because Epcam positivity is probably the best established method to isolate BECs by flow cytometry [8,14].

Other frequently used surface markers to isolate BECs, such as PROM1 (aka CD133) and TROP2 are enriched in most BEC populations as well. Although TROP2 is described to be only expressed in mouse activated BECs, we see high enrichment of this gene in Hnf1β and Mic1c3 isolated BECs from healthy mouse livers, suggesting that TROP2 is also expressed in quiescent BECs (Figure 2). This confirms a previous study, in which TROP2 protein is indeed expressed in both quiescent and activated BECs in human livers from ASH patients [25].

Less popular methods to isolate BECs are functional assays, such as the side population (SP) and aldehyde dehydrogenase (ALDH) activity [20,21]. SP is based on efflux of Hoechst by ABC transporters, while ALDH activity assays rely on the conversion of a fluorescent molecule into a negatively charged dye initiated by ALDH enzymes [21,38]. Although these functional assays rely on protein activity, gene expression levels of both Abcc1 and Aldh1a2 are likewise enriched in human and mouse BECs (Figure 2).

### 3.3. Pathway Analysis Reveals New Potential Pathway in BEC Activation: KRAS Signalling

Single gene expression analysis of BEC transcriptomic profiles is a useful tool to look for biomarkers but cannot gain much insight into the function of BECs in healthy and damaged livers. Pathway analysis, on the other hand, looks at gene sets and can predict biological functions of a cell type. We used gene set enrichment analysis (GSEA) on all human and mouse BECs and made a distinction between mouse BEC cell lines, quiescent and activated BECs, and BECs isolated after different days of DDC diet (day 0, 3 and 7). 

Gene expression profiles of BEC cell lines are obviously enriched in pathways that are involved in cell growth and proliferation, such as mTor and Myc signaling, mitotic spindle and G2M checkpoint control (Figure 3). Most quiescent BEC populations express genes involved in Hedgehog, Notch- and TGFβ-signaling, all pathways previously described to control the BEC phenotype [39,40,41,42,43]. Strikingly, TNF-signaling is the only pathway that is enriched in all BEC data sets analyzed and the KRAS signaling pathway seems to be only enriched in activated BEC populations (Figure 3). Note that not all pathways are enriched in both activated human and mouse BECs.

### 3.4. Creation of a Unique Quiescent and Activated BEC Gene Signature

BECs isolated using different markers have a high variety in enriched genes and pathways. Therefore, we wanted to create a unique BEC gene set that can recognize BECs, mouse or human, in any conditions. To discard genes that are effected by batch effects, we used a fold change of at least 8 times or higher since most commonly used BEC genes are at least enriched 8 times (Figure 1D). We found 417 genes that were highly enriched in quiescent BECs, when comparing healthy livers, and identified 301 genes, when comparing BEC gene profiles, to the average expression of other cell types, such as immune cells, quiescent and activated stellate cells, endothelial cells, epithelial cells and hepatocytes (Figure 4A,B). By merging both gene sets, we created a mouse quiescent BEC signature containing 205 genes (Figure 5A). Our aim was to create a gene signature for both mouse and human BECs so we compared our mouse signature with our previously published human BEC signature [25]. Finally, this resulted in a signature for quiescent human and mouse BECs consisting of 50 genes predominantly involved in cell growth, extracellular matrix organization and morphogenesis (Table 3, Figure 5B). Using the same strategy, we obtained 725 genes enriched in activated mouse BECs compared to healthy livers and 628 genes when compared to different cell types (Figure 4C,D). In this comparison, we did not include Lrg5+ gene profiles since all the other mouse and human BEC populations are not enriched in LGR5 levels and the Lrg5+ population does not express the majority of typical BEC markers. When merging both gene sets (490 genes) with the human BEC signature, we obtained an activated mouse and human BEC signature of 83 genes that are mainly involved in extracellular matrix organization and tissue development (Figure 5A,C, Table 3). Thus, we generated a quiescent and activated BEC signature containing 50 and 83 genes, respectively. Interestingly, 39 genes are present in both signatures suggesting that these are “bona fide” BEC genes and can be used to identify both quiescent and activated BECs in human and mouse.

### 3.5. BEC Gene Signatures As a Tool for Identification of BECs

Gene signatures are frequently used to study functions of cells, but also to identify specific populations. Therefore, we tested our signature on BEC-like cells and compared it to two manually established BEC signatures [30,44]. We found that only our signatures were highly enriched in undifferentiated HepaRGs (human BEC stem cell line [45]) when compared to hepatocytes and in fresh or cultured BECs differentiated from induced pluripotent stem cells (iPSCs) when compared to iPSCs. Note that our signatures were not, or poorly, enriched in fetal hepatocytes (versus primary hepatocytes) and differentiated HepaRGs (Figure 6A).

BECs are known to be activated in many human liver diseases, such as alcoholic steatohepatitis and cirrhosis [44,46]. We therefore compared all BEC signatures to evaluate BEC enrichment in different human liver disorders. Our results show that only our generated signatures and the previously reported signature of Sancho-Bru et al. [44] are able to detect BEC enrichment in alcoholic hepatitis, NASH and peribiliary sclerotic livers. Remarkably, none of the signatures can recognize BECs in peribiliary cirrhosis (PBC) livers, even though it is well known that this disease state is accompanied with activated BECs (Figure 6B).

Recently, Pepe-Mooney et al. performed single cell RNA sequencing (scRNAseq, [47]) on BECs (Epcam^+^) and hepatocytes isolated from healthy and DDC-injured livers [27]. Unbiased detection of different cell populations are performed by tSNE plot and verified using well-known markers. We used their single cell transcriptome data and created a tSNE plot to verify our BEC signatures. This plot clearly presents four different groups (healthy and DDC-injured hepatocytes and BECs), which was confirmed by commonly used BEC and hepatocyte markers (Appendix A). Almost all hepatocytes expressed Cyp3a11, Cyp2e1 and albumin while the BEC fraction expressed Sox9 and Epcam. Tacstd2 (aka Trop2) was expressed only in BECs isolated from DDC-injured livers, suggesting that this fraction contains activated BECs. Instead of validating cell populations using one marker, Pont et al. designed “Single Cell Signature explorer” to calculate a signature score at the single cell level and visualize these scores on tSNE plots [48]. Using this tool, we calculated our activation and quiescent BEC signatures on scRNA seq data and represented them in tSNE plots (Appendix A). Our results show that our quiescent signature was enriched in most BECs isolated from healthy livers, while our activation signature was scattered between both BECs from healthy or DDC injured mice. This suggests that these signatures are not sufficiently restricted to distinguish these populations in this scRNAseq data set that only consists of BECs and hepatocytes. To establish a scRNAseq signature for BECs, we extracted genes that are mainly enriched in BECs using scRNAseq data of Aizarani et al., 2019 [28]. This data set contains single cell expression of BECs, hepatocytes, liver sinusoidal endothelial cells (LSECs), Kupffer cells and other immune cells, such as natural killer cells and cytotoxic T-cells (Appendix A). We first extracted genes that are only expressed in human BECs with at least an average fold change of 2. By merging these genes with our quiescent and activated signature, we created a scBEC signature containing 9 genes (SOX9, MAGI1, BICC1, EPCAM, SLC5A1, DCDC2, ITGB8, KRT7 and CYR61, Figure 5D, Figure 7A). Next, we validated this scBEC signature in the tSNE plot of Pepe-Mooney et al. and confirmed that almost all genes were highly enriched in only quiescent as well as activated mouse BECs (Figure 7B). Note that mouse BECs and hepatocytes do not express Itgb8.

## 4. Discussion

BECs are intensively studied due to their potential to proliferate and differentiate into hepatocytes and cholangiocytes in vivo and in vitro. They form a type of backup system which is activated when hepatocytes no longer have the capacity to restore the liver cell mass [49,50]. Many studies describe the isolation of BECs from healthy and injured mouse livers and performed transcriptomic analysis to gain more insight into their function and their regulation. However, from these studies it was never clear whether the different reports were actually looking at the same BEC population since different isolation techniques and liver injury models were used. We therefore compared transcriptome data of BECs isolated from different injury models using different approaches. By doing so we established gene signatures that can be used to detect the presence or enrichment of BECs in gene expression data sets obtain through either standard gene array platforms or from (sc)RNA sequencing. A final scBEC signature, which consists of BEC-related and unrelated genes, was very efficient in separating qBECs and aBECs from other liver cell types in tSNE plots obtained from scRNAseq data.

Until recently, it was not common practice to compare the gene profile data of the cell type that one wanted to report on, to gene expression data of a similar cell type reported previously by someone else. Often the data had not been deposited in a public database or there has been an embargo period to download the data. This way researchers withhold published data from direct competitors or those with contrary views. It is fortunate that this trend is changing, and we used the kindly shared data (Table 1) to compare the BEC gene profiles with the aim to generate a gene signature for BECs and as a result of this perhaps identify novel genes involved in BEC biology. 

We found that microarray platforms and the method used to isolate the BECs have a high impact on the transcriptome data obtained. However, most BEC populations are still highly enriched in popular BEC markers such as Krt19, Epcam and Sox9, but are clearly not similar and thus could affect the conclusions drawn in these studies. The difference in gene profiles can also be explained by the fact that the isolation procedures used were all based on the enrichment by only one BEC-marker (with exception of Mic1c3). Indeed, our results suggest that isolation of BECs based on only Epcam is not a good option, because this population is poorly enriched in Sox9, Hnf1β and Abcc1. A reason for this could be that these represent rarer subtypes of BECs are more difficult to detect in bulk transcriptomic data of Epcam positive cells. However, since gene expression of Epcam itself is highly enriched in all BEC populations, we do believe that Epcam-based cell sorting is a good way to enrich for BECs, but in combination with other markers. This corroborates studies by Lu et al. who investigated multiple markers for the isolation of BECs and found that CD45^-^CD31^-^Ter119^-^Epcam^+^CD24^+^CD133^+^ cells are true biliary cells, unfortunately the authors did not perform extensive gene expression profiling [14]. 

The BEC transcriptomic data obtained from human and mouse livers clearly show differences. Appendix A shows a list of genes only expressed in human or mouse BECs which suggest that there are some differences between human and mouse BECs. Importantly, several papers have shown that analyzing transcriptome using RNA-seq and microarray can already result in significant differences [51,52,53]. Microarray depends on the variety of probes that can detect specific RNA molecules while RNA-seq analyses all RNAs. RNA-seq is therefore more sensitive and will find more differentially regulated genes which can result in finding different pathways. There is a platform-specific batch effect which cannot be circumvented since all quiescent mouse BECs were analyzed using different platforms. To overrule this batch effect, we included only genes in our BEC gene signatures that showed a >8 fold enrichment over healthy livers.

One could argue that, with the introduction of scRNAseq analysis, markers for BECs are not needed anymore to carry out transcriptome profiling of BECs in healthy and diseased states. However, due to the abundance of hepatocytes, macrophages and endothelial cells in a single cell suspension of livers, one still needs to enrich for low abundance cell types to obtain sufficient cells to be sequenced. A recent publication investigated human BECs on single cell level and reported the isolation of EPCAM^+^CDH6^+^TROP2- BECs [54]. CDH6 is also present in our BEC gene signature, and it is clear from the scRNAseq data from Pepe-Mooney et al., 2019 [27] (Figure 5C) that a negative selection for TROP2 would not discard many cells from an EPCAM isolation. However, by a negative selection of TROP2 one might dispose of a separate BEC population since TROP2 positive cells from alcoholic steatohepatitis livers are enriched in most BEC markers and TROP2 itself is highly enriched in almost all human and mouse BEC transcriptomes analyzed (Figure 2 and [19]). 

In our efforts to identify additional BEC markers, we reviewed the expression patterns of our signature genes using the human protein atlas (www.proteinatlas.org). Several proteins from our signatures, previously not associated with BECs, are localized to biliary epithelial cells or ductular reactions (C1orf116, CDH6, S100A6, MYO6, AKAP7 and DCDC2) and thus can be used to identify or isolate (subsets) of BEC populations (Appendix A). Their localizations in the liver and their association with an BEC signature makes these proteins interesting subjects for further analysis in BEC biology. ScRNA sequencing provides much more information compared to bulk microarray or RNA sequencing because the transcriptome from every cell is analyzed separately. Still, it does not necessarily give insight into the protein production and localization of the gene of interest in the liver. A recent study combined scRNA sequencing with RNA hybridization of livers, an approach that could be used to look for RNA as a marker for BECs instead of proteins [55]. Cellular indexing of transcriptomes and epitopes by sequencing (CITE-seq) is another recent development in which transcriptome analysis is combined with expression of a panel of proteins at a single cell level [56]. With this strategy, gene expression can be related to protein expression and could perhaps lead to the identification of other membrane proteins that are strongly linked to an BEC gene profile. Of course, in the case of BECs, one would still need to enrich for an BEC-like cell before CITE-seq analysis due to the low abundance of these cells in the liver.

Merging differently isolated BEC populations can remove batch effects and therefore be more precise when performing pathway analysis. We found that the KRAS signaling pathway is enriched in BECs that are being activated that thus could be important for BEC activation. Recent studies reported that KRAS signaling is indeed important in self-renewal, proliferation and differentiation of hematopoietic and induced pluripotent stem cells [57,58,59], however further studies are needed to determine whether this pathway is also essential for BEC biology.

Gene signatures can be used to identify certain isolated cell populations. We created a quiescent and activated BEC gene signature that can recognize BEC-like cells and detect BECs in liver diseases accompanied with an activation of the BEC compartment. Interestingly, both the quiescent and activated BEC signature were equally enriched in PSC and NASH, suggests that both populations are present. There is less gene enrichment for AH and AC, which suggests less BEC expansion in these diseases or a higher variability in AH and AC. Strangely, we found no enrichment in PBC which is known to contain expanded BECs [25]. We also investigated whether genes of the scBEC signature could be used to examine for BEC activation in the different liver diseases. Only EPCAM, SOX9 and DCDC2 were enriched in ASH and Cirrhotic livers (Appendix A). This indicates that the scBEC signature is not able to identify activated BECs in bulk transcriptomic data of diseased livers. Probably because this scBEC signature only recognizes certain subsets of activated BECs while the quiescent and activated signatures can identify higher variety of BECs.

ScRNA seq is becoming the standard when analyzing transcriptomes of specific cell population. Several recent publications performed scRNA seq to study mechanisms of BECs from injured livers [27,60]. One of the main issues of scRNA seq is to categorize different cell clusters within a t-SNE plot. Most researchers use multiple gene markers to identify these clusters. We found that our gene signature, created by bulk microarray and RNA seq, was not able to recognize all activated BECs in a t-SNE plot representing hepatocytes and Epcam positive cells isolated from healthy and diseased mouse livers. This suggests that gene signatures might need to be re-evaluated, depending on the complexity of the scRNA sequencing data. In the example described here, we found several genes from our quiescent and activation gene signature that were only expressed in human BECs, based on tSNE plot from the data of Aizarani et al., 2019 [28]. However, these few genes might not be specific enough if other cell types such as hepatic stellate cells cells are included into the tSNE plot. For instance, S100A6 was recently identified as a universal marker of activated myofibroblasts [61]. Future scRNAseq data sets representing equal amounts of different liver cell populations in one tSNE plot will determine whether the BEC signatures that we describe here will be sufficient to identify quiescent and activated BECs in mouse or human livers. 

To summarize, we describe here for, the first time, a meta-analysis on gene profiles obtained from BECs isolated from different liver injury models and human diseased livers, and analyzed by several different analysis platforms. We created a unique gene signature for the identification of BECs in bulk microarray and RNAseq data sets, but also created a gene signature that identifies both quiescent and activated BECs in a scRNA seq data set. 

## Figures and Tables

**Figure 1 cells-08-01117-f001:**
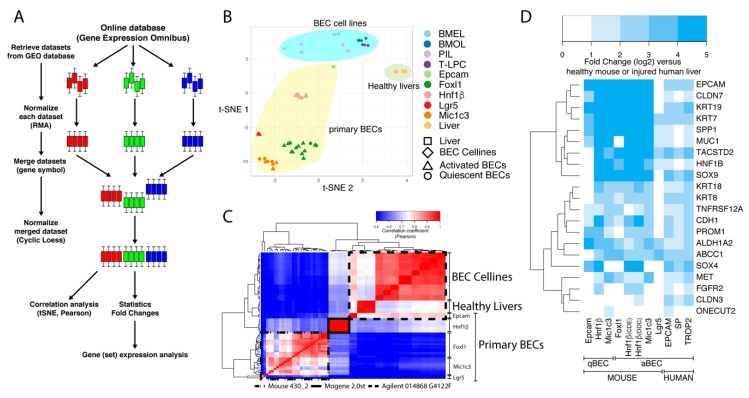
Workflow and clustering of BEC transcriptome data sets. (**A**) Schematic overview of the workflow to merge microarray data from different platforms. Colored bars represent boxplots of the expression of all genes from one sample. Three colors are an example of three different datasets with multiple samples. (**B**,**C**) T-distributed stochastic neighbor embedding (t-SNE) and Pearson correlation analysis of mouse transcriptomic data from BEC cell lines, primary BECs and healthy livers. (**D**) Relative gene expression analysis (fold change, log2) of common BEC markers in primary human and mouse BECs compared to livers.

**Figure 2 cells-08-01117-f002:**
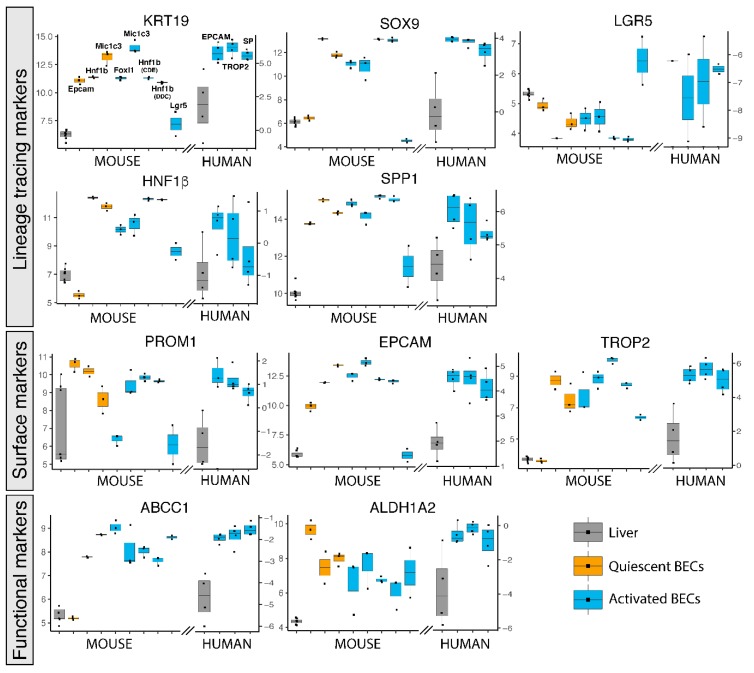
Gene expression of popularly used BEC markers for isolation in primary human and mouse BECs. Gene expression of mouse and human livers (grey), quiescent (orange) and activated (blue) BECs. Left axis represents normalized intensity values (log2) for all mouse gene expression and right axis normalized transcript per million (log2) for human data.

**Figure 3 cells-08-01117-f003:**
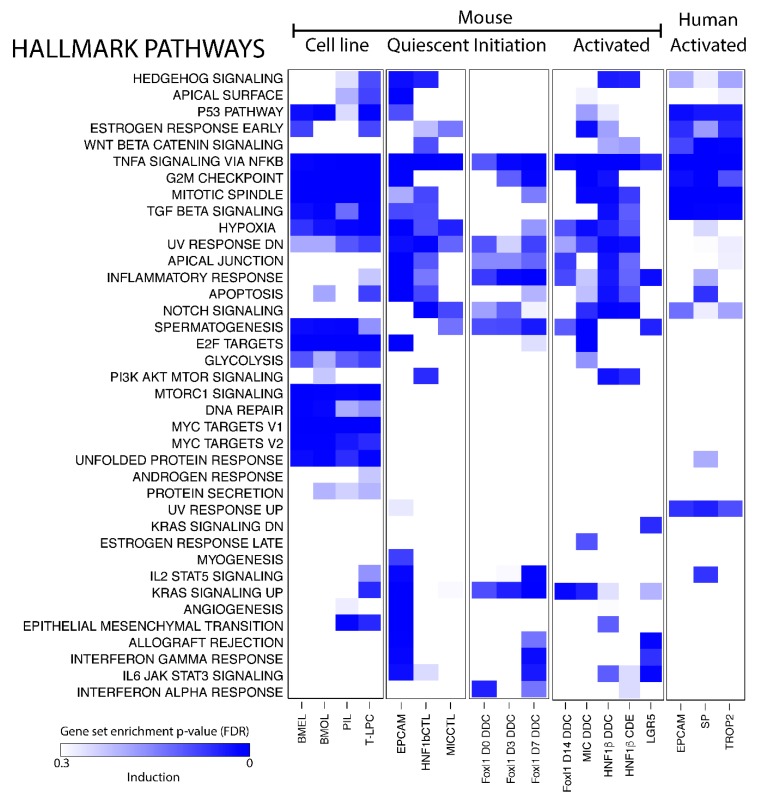
Pathway analysis of BEC populations. Gene set enrichment analysis of Hallmark pathways on transcriptomic profiles of BECs compared to healthy (mouse) or injured (human) livers. Gradient of the blue color represents positively enriched pathway (False discovery rate, FDR).

**Figure 4 cells-08-01117-f004:**
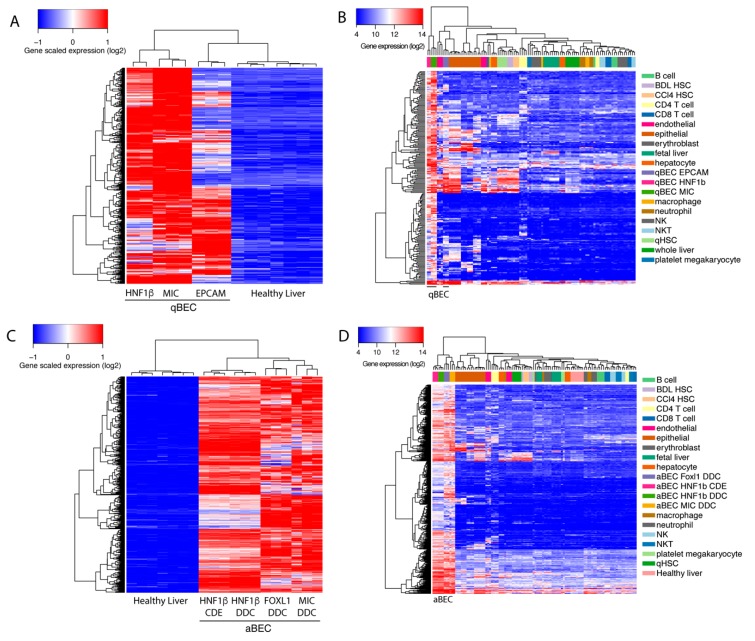
Selection of genes enriched in BECs. Heatmap of genes enriched in BECs compared to healthy livers (**A** and **C**) or different cell types (**B** and **D**). For A only genes enriched in BECs isolated from healthy mouse livers were used while for C only genes enriched in BECs isolated from either choline-deficient, ethionine-supplemented (CDE)- or 3,5-diethoxycarbonyl-1,4-dihydrocollidine (DDC)-treated mice.

**Figure 5 cells-08-01117-f005:**
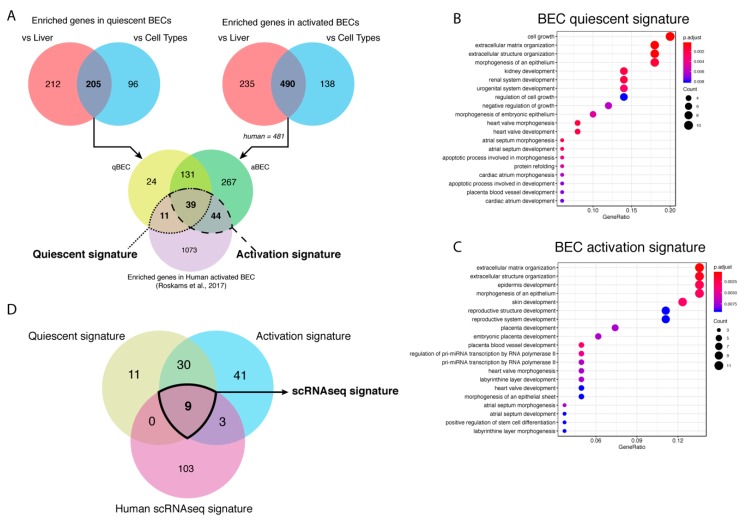
Generation of BEC gene signatures. (**A**) Venn diagrams of the genes enriched in BECs compared to healthy livers or different cell types and both merged gene sets compared to genes enriched in human BECs. Gene ontology (GO) analysis of quiescent (**B**) and activated (**C**) BEC gene signature. The size of the circle is correlated to the number of genes involved in that GO. The color of the circle represents significance (adjusted *p*-value) and the *x*-axis stands for gene ratio (genes within signature versus total number of genes in GO). (**D**) Venn diagram of genes enriched in quiescent and activated BECs (microarray/RNAseq) and in BECs from scRNAseq of human livers.

**Figure 6 cells-08-01117-f006:**
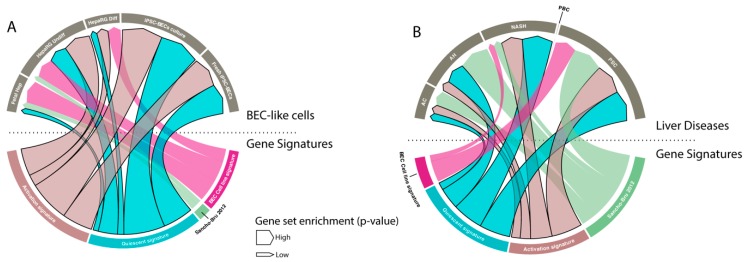
BEC signatures to identify BECs. (**A** and **B**) Visual representation (chord diagram) of gene set enrichment analysis with our quiescent and activated BEC signatures, Sancho-Bru signature (2012) [44] and BEC cell line signature of Passman et al. 2016 [30] on BEC-like cells and liver diseases (AC: alcoholic cirrhosis, AH: alcoholic hepatitis, NASH: non-alcoholic steatohepatitis, PBC: peribiliary cirrhosis, PSC: primary sclerosis cholangitis). The size of the arrow presents the positive enrichment (significance, -log *p*-value).

**Figure 7 cells-08-01117-f007:**
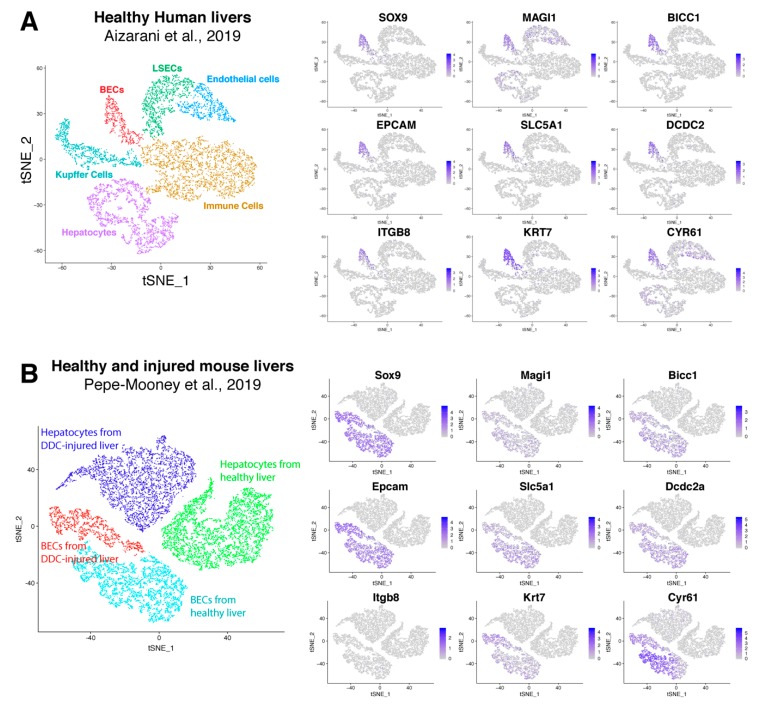
Validation of scBEC signature. TSNE plots of human (**A**) and mouse (**B**) liver cells from respectively Aizarani et al. 2019 [28] and Pepe-Mooney et al. 2019 [27], with expression of every single gene of the scBEC signature. Blue scale bars represent normalized counts.

**Table 1 cells-08-01117-t001:** Biliary epithelial cells (BEC) gene expression data.

LPC Marker	Reference	Healthy			Injury			Hepatocytes	Niche	Injury Model	Species	Platform	GSEA
		LPC	Liver	Negative Fraction	LPC	Liver	Negative Fraction						
LGR5	[23]		x		x	x		x		CCl4	Mouse	Agilent 014868 G4122F	GSE32210
MIC1 1C3	[18]	x		x	x		x			DDC	Mouse	Agilent 014868 G4122F	GSE29121
HNF1b	[10]	x		x	x					DDC / CDE	Mouse	Mogene 2.0st	GSE51389
Foxl1	[29]			x	x		x			DDC	Mouse	Agilent 014868 G4122F	GSE28892
LPC cell lines	[30]	x	x							None	Mouse	Mouse 430_2	GSE85114/GSE12908/GSE18269
Side population	[25]				x	x			x	ASH	Human	Truseq LT	GSE102683
EpCAM	[25]				x	x			x	ASH	Human	Truseq LT	GSE102683
TROP2	[25]				x	x			x	ASH	Human	Truseq LT	GSE102683
EpCAM	[24]	x								None	Mouse	Mouse 430_2	GSE63793

**Table 2 cells-08-01117-t002:** Microarray GEO sample (GSM) number associated with cell type or tissue.

GSM Number	Cell Type or Tissue	Specie
GSM1061907, GSM1061908, GSM686644, GSM686645	Bcell	Mouse
GSM1071644, GSM361592, GSM389821	platelet_megakaryocyte	Mouse
GSM1081398 - GSM1081400	CD8_T_cell	Mouse
GSM1129665, GSM686658, GSM686659	macrophage	Mouse
GSM1214478-GSM1214480, GSM686650, GSM686651	NK	Mouse
GSM1232678 - GSM1232680, GSM686652, GSM686653	NKT	Mouse
GSM1281320 - GSM1281322, GSM525382 - GSM525384	hepatocyte	Mouse
GSM1301661, GSM1301662	neutrophil	Mouse
GSM190795 - GSM190797, GSM598999 - GSM599002, GSM658893 - GSM658898, GSM765922 - GSM765924	epithelial	Mouse
GSM216494 - GSM216497, GSM686654	erythroblast	Mouse
GSM298115 - GSM298117, GSM378250 - GSM378254	fetal_liver	Mouse
GSM344315 - GSM344318	neutrophil	Mouse
GSM547762 - GSM547766, GSM690763 - GSM690765	endothelial	Mouse
GSM555381 - GSM555384	CD4_T_cell	Mouse
GSM571897	macrophage	Mouse
GSM591473, GSM591475, GSM591477, GSM591480, GSM602665 - GSM602667	Healthy Liver	Mouse
GSM686646, GSM686647	CD4_T_cell	Mouse
GSM686648, GSM686649	CD8_T_cell	Mouse
GSM852330 - GSM852334	Quiescent HSC	Mouse
GSM852341 - GSM852343	HSC from BDL	Mouse
GSM852344 - GSM852346	CCl4_HSC	Mouse
GSM1557526 - GSM1557528	LPC EPCAM	Mouse
GSM2257924 - GSM2257940	PIL	Mouse
GSM2257941 - GSM2257944	BMOL	Mouse
GSM2257945 - GSM2257947	T_LPC	Mouse
GSM323977 - GSM323981	BMEL	Mouse
GSM715841 - GSM715844	LPC_Foxl1_D0	Mouse
GSM715856 - GSM715859	LPC_Foxl1_POS_D3	Mouse
GSM715860 - GSM715863	LPC_Foxl1_POS_D7	Mouse
GSM715864 - GSM715866	LPC_Foxl1_POS_D14	Mouse
GSM715867 - GSM715870	LPC_Foxl1_undiff	Mouse
GSM721145 - GSM721149	LPC_MIC_POS_CTL	Mouse
GSM721153 - GSM721156	LPC_MIC_POS_DDC	Mouse
GSM1047599, GSM1047603	LPC_LGR5	Mouse
GSM1244523, GSM1244525	LPC_HNF1b_POS_CDE	Mouse
GSM1244526 - GSM1244528	LPC_HNF1b_POS_DDC	Mouse
GSM1244529, GSM1244530	LPC_HNF1b_POS_CTL	Mouse
GSM709348 - GSM709354	Healthy Livers for ASH	Human
GSM709355 - GSM709369	ASH	Human
GSM1974233, GSM1974234	Primary hepatocytes	Human
GSM1974235, GSM1974236	Fetal hepatocytes	Human
GSM1627740 - GSM1627773	Healthy livers for NASH	Human
GSM1627805, GSM1627806	Definite NASH	Human
GSM155919, GSM155926 - GSM155928, GSM155947, GSM155948, GSM155961, GSM155964, GSM155988, GSM155989	Healthy livers as control for Cirrhotic livers	Human
GSM155920 - GSM155923, GSM155931, GSM155951, GSM155952, GSM155965 - GSM155969, GSM155984	Cirrhotic livers	Human
GSM2787428, GSM2787427	Human iPSC	Human
GSM2787426, GSM2787425	Cultured iPSC-LPC	Human
GSM2787422, GSM2787421	Fresh iPSC-LPC	Human
GSM456340 - GSM456342	HepaRG_diff	Human
GSM456343 - GSM456345	HepaRG_undiff	Human
GSM456349 - GSM456351	Primary hepatocytes Control for HepaRG	Human

**Table 3 cells-08-01117-t003:** List of human and mouse genes in quiescent and activated BECs.

HUMAN				MOUSE		HUMAN/MOUSE
Gene Name	Gene Description	Entrez ID	Ensemble ID	Gene Name	Ensemble ID	LPC expression
AKAP7	A-kinase anchoring protein 7	9465	ENSG00000118507	Akap7	ENSMUSG00000039166	Activated
ALDH1A2	aldehyde dehydrogenase 1 family member A2	8854	ENSG00000128918	Aldh1a2	ENSMUSG00000013584	Quiescent
ANKRD1	ankyrin repeat domain 1	27063	ENSG00000148677	Ankrd1	ENSMUSG00000024803	Activated
ANKRD42	ankyrin repeat domain 42	338699	ENSG00000137494	Ankrd42	ENSMUSG00000041343	Activated
ARL14	ADP ribosylation factor like GTPase 14	80117	ENSG00000179674	Arl14	ENSMUSG00000098207	Quiescent and Activated
ATF3	activating transcription factor 3	467	ENSG00000162772	Atf3	ENSMUSG00000026628	Quiescent
B4GALT5	beta-1,4-galactosyltransferase 5	9334	ENSG00000158470	B4galt5	ENSMUSG00000017929	Activated
BICC1	BicC family RNA binding protein 1	80114	ENSG00000122870	Bicc1	ENSMUSG00000014329	Quiescent and Activated
C1orf116	chromosome 1 open reading frame 116	79098	ENSG00000182795	AA986860	ENSMUSG00000042510	Quiescent and Activated
CDH1	cadherin 1	999	ENSG00000039068	Cdh1	ENSMUSG00000000303	Activated
CDH6	cadherin 6	1004	ENSG00000113361	Cdh6	ENSMUSG00000039385	Activated
CREB5	cAMP responsive element binding protein 5	9586	ENSG00000146592	Creb5	ENSMUSG00000053007	Activated
CRYAB	crystallin alpha B	1410	ENSG00000109846	Cryab	ENSMUSG00000032060	Quiescent and Activated
CTGF	connective tissue growth factor	1490	ENSG00000118523	Ctgf	ENSMUSG00000019997	Quiescent and Activated
CYR61	cysteine rich angiogenic inducer 61	3491	ENSG00000142871	Cyr61	ENSMUSG00000028195	Quiescent and Activated
DCDC2	doublecortin domain containing 2	51473	ENSG00000146038	Dcdc2a	ENSMUSG00000035910	Quiescent and Activated
DDR1	discoidin domain receptor tyrosine kinase 1	780	ENSG00000223680	Ddr1	ENSMUSG00000003534	Activated
DSP	desmoplakin	1832	ENSG00000096696	Dsp	ENSMUSG00000054889	Activated
EGR2	early growth response 2	1959	ENSG00000122877	Egr2	ENSMUSG00000037868	Quiescent
EHF	ETS homologous factor	26298	ENSG00000135373	Ehf	ENSMUSG00000012350	Quiescent and Activated
ELOVL7	ELOVL fatty acid elongase 7	79993	ENSG00000164181	Elovl7	ENSMUSG00000021696	Activated
ENC1	ectodermal-neural cortex 1	8507	ENSG00000171617	Enc1	ENSMUSG00000041773	Activated
ENTPD2	ectonucleoside triphosphate diphosphohydrolase 2	954	ENSG00000054179	Entpd2	ENSMUSG00000015085	Quiescent
EPCAM	epithelial cell adhesion molecule	4072	ENSG00000119888	Epcam	ENSMUSG00000045394	Quiescent and Activated
FBRS	fibrosin	64319	ENSG00000156860	Fbrs	ENSMUSG00000042423	Activated
FLRT3	fibronectin leucine rich transmembrane protein 3	23767	ENSG00000125848	Flrt3	ENSMUSG00000051379	Quiescent and Activated
FOSB	FosB proto-oncogene, AP-1 transcription factor subunit	2354	ENSG00000125740	Fosb	ENSMUSG00000003545	Quiescent and Activated
FOXJ1	forkhead box J1	2302	ENSG00000129654	Foxj1	ENSMUSG00000034227	Activated
FRAS1	Fraser extracellular matrix complex subunit 1	80144	ENSG00000138759	Fras1	ENSMUSG00000034687	Quiescent
GADD45B	growth arrest and DNA damage inducible beta	4616	ENSG00000099860	Gadd45b	ENSMUSG00000015312	Activated
GLIS2	GLIS family zinc finger 2	84662	ENSG00000274636	Glis2	ENSMUSG00000014303	Quiescent and Activated
GLIS3	GLIS family zinc finger 3	169792	ENSG00000107249	Glis3	ENSMUSG00000052942	Quiescent and Activated
GOLGB1	golgin B1	2804	ENSG00000173230	Golgb1	ENSMUSG00000034243	Quiescent
HBEGF	heparin binding EGF like growth factor	1839	ENSG00000113070	Hbegf	ENSMUSG00000024486	Quiescent and Activated
HSPA1A	heat shock protein family A (Hsp70) member 1A	3303	ENSG00000237724	Hspa1a	ENSMUSG00000091971	Quiescent
HSPA1B	heat shock protein family A (Hsp70) member 1B	3304	ENSG00000224501	Hspa1b	ENSMUSG00000090877	Quiescent
ITGB4	integrin subunit beta 4	3691	ENSG00000132470	Itgb4	ENSMUSG00000020758	Activated
ITGB8	integrin subunit beta 8	3696	ENSG00000105855	Itgb8	ENSMUSG00000025321	Quiescent and Activated
JUNB	JunB proto-oncogene, AP-1 transcription factor subunit	3726	ENSG00000171223	Junb	ENSMUSG00000052837	Activated
JUND	JunD proto-oncogene, AP-1 transcription factor subunit	3727	ENSG00000130522	Jund	ENSMUSG00000071076	Quiescent and Activated
KIAA1324	KIAA1324	57535	ENSG00000116299	5330417C22Rik	ENSMUSG00000040412	Quiescent and Activated
KLF5	Kruppel like factor 5	688	ENSG00000102554	Klf5	ENSMUSG00000005148	Quiescent and Activated
KRT17	keratin 17	3872	ENSG00000128422	Krt17	ENSMUSG00000035557	Activated
KRT19	keratin 19	3880	ENSG00000171345	Krt19	ENSMUSG00000020911	Quiescent and Activated
KRT7	keratin 7	3855	ENSG00000135480	Krt7	ENSMUSG00000023039	Quiescent and Activated
LAMB2	laminin subunit beta 2	3913	ENSG00000172037	Lamb2	ENSMUSG00000052911	Quiescent
LAMC2	laminin subunit gamma 2	3918	ENSG00000058085	Lamc2	ENSMUSG00000026479	Quiescent and Activated
LRRC49	leucine rich repeat containing 49	54839	ENSG00000137821	Lrrc49	ENSMUSG00000047766	Activated
MACC1	MET transcriptional regulator MACC1	346389	ENSG00000183742	Macc1	ENSMUSG00000041886	Quiescent and Activated
MAGI1	membrane associated guanylate kinase, WW and PDZ domain containing 1	9223	ENSG00000151276	Magi1	ENSMUSG00000045095	Quiescent and Activated
MYO5C	myosin VC	55930	ENSG00000128833	Myo5c	ENSMUSG00000033590	Activated
MYO6	myosin VI	4646	ENSG00000196586	Myo6	ENSMUSG00000033577	Activated
NFAT5	nuclear factor of activated T cells 5	10725	ENSG00000102908	Nfat5	ENSMUSG00000003847	Activated
NFE2L3	nuclear factor, erythroid 2 like 3	9603	ENSG00000050344	Nfe2l3	ENSMUSG00000029832	Quiescent
NFKBIE	NFKB inhibitor epsilon	4794	ENSG00000146232	Nfkbie	ENSMUSG00000023947	Activated
NOTCH2	notch 2	4853	ENSG00000134250	Notch2	ENSMUSG00000027878	Quiescent and Activated
NSD1	nuclear receptor binding SET domain protein 1	64324	ENSG00000165671	Nsd1	ENSMUSG00000021488	Quiescent and Activated
PEG10	paternally expressed 10	23089	ENSG00000242265	Peg10	ENSMUSG00000092035	Activated
POGZ	pogo transposable element derived with ZNF domain	23126	ENSG00000143442	Pogz	ENSMUSG00000038902	Activated
PPP1R9A	protein phosphatase 1 regulatory subunit 9A	55607	ENSG00000158528	Ppp1r9a	ENSMUSG00000032827	Activated
RAI2	retinoic acid induced 2	10742	ENSG00000131831	Rai2	ENSMUSG00000043518	Quiescent and Activated
RASSF9	Ras association domain family member 9	9182	ENSG00000198774	Rassf9	ENSMUSG00000044921	Quiescent and Activated
RBM25	RNA binding motif protein 25	58517	ENSG00000119707	Rbm25	ENSMUSG00000010608	Activated
RIPK4	receptor interacting serine/threonine kinase 4	54101	ENSG00000183421	Ripk4	ENSMUSG00000005251	Activated
S100A6	S100 calcium binding protein A6	6277	ENSG00000197956	S100a6	ENSMUSG00000001025	Quiescent and Activated
SERPINH1	serpin family H member 1	871	ENSG00000149257	Serpinh1	ENSMUSG00000070436	Quiescent
SF1	splicing factor 1	7536	ENSG00000168066	Sf1	ENSMUSG00000024949	Activated
SHROOM3	shroom family member 3	57619	ENSG00000138771	Shroom3	ENSMUSG00000029381	Quiescent and Activated
SLC5A1	solute carrier family 5 member 1	6523	ENSG00000100170	Slc5a1	ENSMUSG00000011034	Quiescent and Activated
SLC6A6	solute carrier family 6 member 6	6533	ENSG00000131389	Slc6a6	ENSMUSG00000030096	Activated
SLC7A1	solute carrier family 7 member 1	6541	ENSG00000139514	Slc7a1	ENSMUSG00000041313	Activated
SLCO3A1	solute carrier organic anion transporter family member 3A1	28232	ENSG00000176463	Slco3a1	ENSMUSG00000025790	Activated
SNRNP200	small nuclear ribonucleoprotein U5 subunit 200	23020	ENSG00000144028	Snrnp200	ENSMUSG00000003660	Activated
SNRNP70	small nuclear ribonucleoprotein U1 subunit 70	6625	ENSG00000104852	Snrnp70	ENSMUSG00000063511	Quiescent and Activated
SOX9	SRY-box 9	6662	ENSG00000125398	Sox9	ENSMUSG00000000567	Quiescent and Activated
SPHK1	sphingosine kinase 1	8877	ENSG00000176170	Sphk1	ENSMUSG00000061878	Activated
SPINT1	serine peptidase inhibitor, Kunitz type 1	6692	ENSG00000166145	Spint1	ENSMUSG00000027315	Quiescent and Activated
SREBF2	sterol regulatory element binding transcription factor 2	6721	ENSG00000198911	Srebf2	ENSMUSG00000022463	Activated
STK35	serine/threonine kinase 35	140901	ENSG00000125834	Stk35	ENSMUSG00000037885	Activated
SYNJ2	synaptojanin 2	8871	ENSG00000078269	Synj2	ENSMUSG00000023805	Quiescent and Activated
TACSTD2	tumor associated calcium signal transducer 2	4070	ENSG00000184292	Tacstd2	ENSMUSG00000051397	Activated
TCF20	transcription factor 20	6942	ENSG00000100207	Tcf20	ENSMUSG00000041852	Activated
TGFB2	transforming growth factor beta 2	7042	ENSG00000092969	Tgfb2	ENSMUSG00000039239	Quiescent and Activated
THSD4	thrombospondin type 1 domain containing 4	79875	ENSG00000187720	Thsd4	ENSMUSG00000032289	Activated
TNFRSF19	TNF receptor superfamily member 19	55504	ENSG00000127863	Tnfrsf19	ENSMUSG00000060548	Activated
TUBB2B	tubulin beta 2B class IIb	347733	ENSG00000137285	Tubb2b	ENSMUSG00000045136	Activated
UBAP2L	ubiquitin associated protein 2 like	9898	ENSG00000143569	Ubap2l	ENSMUSG00000042520	Quiescent and Activated
UGT8	UDP glycosyltransferase 8	7368	ENSG00000174607	Ugt8a	ENSMUSG00000032854	Quiescent and Activated
UNC119B	unc-119 lipid binding chaperone B	84747	ENSG00000175970	Unc119b	ENSMUSG00000046562	Activated
VTCN1	V-set domain containing T cell activation inhibitor 1	79679	ENSG00000134258	Vtcn1	ENSMUSG00000051076	Activated
WFDC2	WAP four-disulfide core domain 2	10406	ENSG00000101443	Wfdc2	ENSMUSG00000017723	Activated
WWC1	WW and C2 domain containing 1	23286	ENSG00000113645	Wwc1	ENSMUSG00000018849	Quiescent and Activated
ZFP36	ZFP36 ring finger protein	7538	ENSG00000128016	Zfp36	ENSMUSG00000044786	Activated
ZFP36L1	ZFP36 ring finger protein like 1	677	ENSG00000185650	Zfp36l1	ENSMUSG00000021127	Activated

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
