# Peer review of "Meta-Analysis of Human and Mouse Biliary Epithelial Cell Gene Profiles"

_cells, 2019, doi:10.3390/cells8101117_

Round 1
Reviewer 1 Report
Authors carried out meta-analysis of gene expression profile data from human and mouse liver progenitor cell to uncover gene sets that are unique to LPC. They concluded that KRAS signaling was new potential pathway in LPC activation. While several findings are interestingly, there are several issues that need to be clarified before publication.
Major Concerns
Selection of LPC dataset How did you select LPC dataset to extract LPC signatures from public database? Arbitrary selection of data source may cause bias in the study. The authors should provide some details in selection procedure for study dataset with rational criteria for inclusion or exclusion as well as specific process like CONSORT diagram.Batch effect There seems to be some issues in batch effect when comparing different data sets together, especially between species, human or mouse. This concern should be addressed.
Estimation of quiescent or activated LPC How did you define the property of LPC regarding quiescent or activated? Figure 1-D is a very tricky example of data clustering from inevitable batch effect. This may not be a strong evidence of sample assignment for LPC property. Each samples should be evaluated carefully in gene ontology or pathway level and then the clustering after merging samples would be possible.
Author Response
Major Concerns
* Selection of LPC dataset How did you select LPC dataset to extract LPC signatures from public database? Arbitrary selection of data source may cause bias in the study. The authors should provide some details in selection procedure for study dataset with rational criteria for inclusion or exclusion as well as specific process like CONSORT diagram.
Indeed, we did not include the criteria of our selection procedure. We now added our procedures in the materials and methods part (“source of gene expression data”) as follows:
“Source of gene expression data. We searched for publicly available transcriptomic data sets of BECs (see table 1). We included only microarray data for mouse BECs with at least two biological repeats and using one of the three most widely used microarray platforms (Agilent 014868, Mouse 430_2 and Mogene 2.0st) to simplify the experiment. We avoided including more microarray platforms to avoid loss of genes due to mismatched annotations between the different platforms.”
* Batch effect There seems to be some issues in batch effect when comparing different data sets together, especially between species, human or mouse. This concern should be addressed.
We agree with the reviewer. Perhaps because we anticipated this, we did not really discuss this in the manuscript. We now added a paragraph in the discussion to address the issue.
“The BEC transcriptomic data obtained from human and mouse livers clearly shows differences. Several papers have shown that analyzing transcriptome using RNA-seq and microarray (resp. human and mouse data in this study) can already result in significant differences [51-53]. Microarray depends on the variety of probes that can detect specific RNA molecules while RNA-seq analyses all RNAs. RNA-seq is therefore more sensitive and will find more differentially regulated genes which can result in the identification of different pathways. There is also a platform-specific batch effect which cannot be circumvented since all quiescent mouse BECs were analyzed using different platforms. To overrule this batch effect we only included genes in our BEC gene signatures that showed a >8 fold enrichment over healthy livers.”
* Estimation of quiescent or activated LPC How did you define the property of LPC regarding quiescent or activated?
This is a valid remark. We assume that quiescent LPCs (now referred to as BECs) are LPCs isolated from healthy livers and activated LPCs from injured livers (DDC diet, CDE diet and CCl4 injury). Of course, when an isolation is carried out on injured livers one will also isolate quiescent LPCs, but the differences in gene expression will have its origin in the activated cells. Since there are no good markers available that can distinguish an activated LPCs from a quiescent LPCs we have to trust the published data sets. These data sets are valuable since we are able to identify gene signatures for LPCs isolated from healthy as well as injured livers, i.e. quiescent and activated LPCs.
* Figure 1-D is a very tricky example of data clustering from inevitable batch effect. This may not be a strong evidence of sample assignment for LPC property. Each samples should be evaluated carefully in gene ontology or pathway level and then the clustering after merging samples would be possible.
We do not entirely understand this remark. Figure 1D is a clustering showing only common LPC markers. It was not our intention to cluster our samples in this figure, merely to illustrate the relative expression of each gene in the different LPC populations.
Reviewer 2 Report
Meta-analysis of human and mouse liver progenitor cell gene profiles
In the present study Verhulst et al collect publicly available transcriptome data sets of human and mouse healthy and injured liver to identify conserved gene signatures of liver progenitor cells (LPCs). The authors focus in particular on the difference in gene expression signatures between activated and quiescent LPCs from bulk transcriptomic data which they validate by overlaying on a single cell study of the liver epithelium before and after injury.
Several recent publications using single cell RNA sequencing have identified new heterogeneity within the biliary populations of both mouse and human liver during homeostasis and disease. Whilst sourced from different tissue sources using different cell collection strategies, many of the findings offer conflicting reports on the phenotype of stable and/or transient progenitor and biliary epithelial cell (BEC) types defined by new and existing markers. Therefore, a study such as this could be greatly beneficial to the field of liver regenerative medicine to identify conserved gene signatures useful for further investigation. Overall the paper is well written and their methodology is clear to follow. However, several serious concerns should be addressed before being considered for publication. In the manuscripts current form, it is unclear how much impact it will have for the field in advancing our understanding of liver progenitor cell biology. Below is a list of concerns that will hopefully improve the quality and impact of the paper.
Major comments
Fundamental to our understanding of liver progenitor cell biology is their relationship with biliary epithelial cells. Much controversy in the field has stemmed from the inability to define a truly a unique LPC signature from mature biliary epithelial cells (BECs). Throughout this study the authors make very little mention of BECs. This should be addressed as many of their transcriptomic data sets (e.g. EpCAM+, TROP2+) likely include BEC populations. For example, in line 192 the authors state ‘KRT19, SOX9, SPP1 and HNF1B are greatly enriched in almost all mouse and human LPC populations confirming that these genes are good candidates to be used as LPC-specific driver genes’. All these genes are highly enriched in BECs (both mouse and human) therefore it is unclear how these genes can be considered LPC specific. Furthermore, the single cell study from Pep Mooney et al 2019 defines the heterogeneity within their biliary population captured by scRNAseq as BECs not progenitor cells. It is therefore misleading to the reader to suggest that the gene signatures defined in this study truly represent LPC signatures and not potential BEC phenotypes.
Several recent single cell human studies provide transcriptomic information of BEC populations. More focus of this study should be on integrated/comparing single cell data sets which may provide a more accurate distinction of potential LPC and BEC phenotypes. In its current format the study is limited by use of Bulk transcriptomic data which does not convince that activated and quiescent signatures can be separated. These studies include MacParland et al 2018 (Ncomms), Aizarani et al 2019 (Nature), Segal et al 2019 (Ncomms) – All human, and Su et al 2017 (BMC Genomics), Planas et al 2019 (Cell Stem Cell) – Mouse. The sequencing data is publicly available for all of these studies and several include non-hepatic cells types for comparison.
The lack of validation of new markers assigned to either activated or quiescent LPCs significantly weakens the impact of this study. While it is understood this study is a meta-analysis I would expect to see the validation of novel identified markers – ideally of the scLPC signature of which only 7 genes were identified. The authors should look for expression, either by IHC, IF or in situ RNA hybridization of newly identified activation markers Slco3a1, Wfdc2 and S100a6 in healthy and DDC-injured liver tissue to validate they are specific to activated LPCs. The authors should also validate the expression of scLPC markers in human cases of AH, AC, NASH and PBC. This would strongly strengthen their claim of distinguishing between activated and quiescent LPCs and increase the confidence in their methodology to identify unique gene signatures. The authors current validation is not sufficient. In supplementary Figure 2 , the authors extract protein atlas staining of novel LPC markers. They do not address that while CDH6, MYO6 and AKAP7 are considered ‘activated markers’ they are enriched in the BECs of non-injured human liver.
In this study the authors use both human and mouse LPC gene profiles analyzed together to formulate a unique signature. This appears counterproductive to the direction of the field. It is now becoming clear that significant differences appear between mouse and human LPC biology. TROP2 for example is highly expressed in activated LPCS in mouse models of liver disease, but highly expressed in the cholangiocytes of non-injured human liver. While it is fine to use mouse LPC data as a framework for identifying unique LPC markers, the authors should include a figure with more in depth comparison of mouse and human signatures to identify key differences and similarities. It will be interesting to observe if similar quiescent/activation signatures will be present when mouse and human transcriptomic data is separated
The authors need to clarify in more detail their labelling of ‘quiescent’ and ‘activated’ LPC. It appears to be based on whether the sample is obtained from healthy or injured liver although this is not clearly stated. In Table 1 there are 3 EpCAM+data sets, 1 mouse and 2 human. 1 EpCAM+human data set contains LPCs from injured liver, but this sample is not present within the tsne plot from Figure 1B (or at least not visible). The three TruSeqLT human data sets from Ceulemans, Verhulst et al 2017 do not have accompanying non-injured healthy liver samples (quiescent). As it is likely EpCAM+ and TROP2+FACS sorting will be obtaining BECs/non activated LPCS, it appears they are assuming they are capturing a mostly activated LPC population. The authors then introduce another category for their pathway analysis, ‘initiation’ (by injury inducing diet length). The authors need to make it much clearer which samples specifically refer to a quiescent or activated state (i.e. healthy or injured) and qualify their reasoning and the limitations for this labelling strategy.
Minor comments
Can the authors explain in their methodology why 8-fold change was used as the cut off for being considered ‘highly enriched’?
In the scRNAseq paper used for validation of the authors quiescent and activated LPC signatures (Pepe-Mooney et al 2019) the gene Cyr61 shows dynamic expression which the authors attribute to fluctuating YAP activity, which is activated upon biliary expansion. Do the authors observe an overlay of their activation LPC gene signature that relates to potential YAP activity (For example – YAP activity in TROP2+cells from DDC injury).
In their introduction the authors should address the difference potential origins of liver progenitor cells in more detail – Hepatocyte dedifferentiationb BEC trans-differentiation or quiescent populations present in healthy tissue with relevant references.
The authors suggest a challenge in the field of studying LPCs is the isolation of LPCs using different markers. EpCAM is the most common marker used in both mouse and human LPC studies (Table 1). Indeed, several recent transcriptomic studies (both mouse and human) use EpCAM+isolation to investigate all biliary cell types – BECs, dedifferentiated hepatocytes and LPCs. Aizarani et al 2019 (Nature), Segal et al 2019 (Ncomms) and Su et al 2017 (BMC Genomics). The authors state themselves that EpCAM is highly expressed in all LPCS populations (line 200), but HNFf1B, Sox9 and Abcc1 are poorly enriched in EpCAM+ This is likely due to these being rarer subtypes that are difficult to detect in bulk transcriptomic analysis. The authors should address this within the discussion (Line 356- 367). The authors should also reword their introduction to (line 72-83), that other LPC isolation techniques (bar Lgr5+) are potentially isolating sub sets of EpCAM+cells, not different populations.
It would be good to see the gene expression patterns of newly identified quiescent and activated markers across the collated data sets (as done for tracing, cell surface and functional markers in figure 2)., to see how closely they represent LPCs isolated from healthy and injured liver respectively. Would a semi-supervised clustering using quiescent and activated gene lists provide better clustering to distinguish quiescent and activated LPC groups not possible in Figure 1B.
Author Response
We thank the reviewer for the kind comments and the thorough reading of the manuscript. We really appreciate all the comments and concerns the reviewer made. We have improved the manuscript considerably according to this reviewers comments.
Major comments
Fundamental to our understanding of liver progenitor cell biology is their relationship with biliary epithelial cells. Much controversy in the field has stemmed from the inability to define a truly a unique LPC signature from mature biliary epithelial cells (BECs). Throughout this study the authors make very little mention of BECs. This should be addressed as many of their transcriptomic data sets (e.g. EpCAM+, TROP2+) likely include BEC populations. For example, in line 192 the authors state ‘KRT19, SOX9, SPP1 and HNF1B are greatly enriched in almost all mouse and human LPC populations confirming that these genes are good candidates to be used as LPC-specific driver genes’. All these genes are highly enriched in BECs (both mouse and human) therefore it is unclear how these genes can be considered LPC specific. Furthermore, the single cell study from Pep Mooney et al 2019 defines the heterogeneity within their biliary population captured by scRNAseq as BECs not progenitor cells. It is therefore misleading to the reader to suggest that the gene signatures defined in this study truly represent LPC signatures and not potential BEC phenotypes.
We agree with the reviewer. In our study, it is difficult to distinguish LPCs from all BECs. We now refer to the cells analyzed as BECs instead of LPCs and have added a sentence in the introduction explaining why we refer to the cells as BECs.
“BECs that can activate are sometimes also referred to as liver progenitor cells. In this manuscript we will refrain from using liver progenitor cells as a term and will refer to quiescent and activated BECs depending on the state of the livers from which the BECs were isolated (healthy vs diseased).”
Several recent single cell human studies provide transcriptomic information of BEC populations. More focus of this study should be on integrated/comparing single cell data sets which may provide a more accurate distinction of potential LPC and BEC phenotypes. In its current format the study is limited by use of Bulk transcriptomic data which does not convince that activated and quiescent signatures can be separated. These studies include MacParland et al 2018 (Ncomms), Aizarani et al 2019 (Nature), Segal et al 2019 (Ncomms) – All human, and Su et al 2017 (BMC Genomics), Planas et al 2019 (Cell Stem Cell) – Mouse. The sequencing data is publicly available for all of these studies and several include non-hepatic cells types for comparison.
Thank you very much for this comment. We agree that comparing scRNAseq data sets is of great value, especially since they most of the time include other cells besides BECs. We agree with the reviewer that our signatures cannot convincingly identify BECs from healthy and injured mice in the Pepe-Mooney et al study. We therefore re-establich a new LPC scRNA seq signature by comparing our quiescent and activated BEC signature with genes expressed in BECs using the data of Aizarani et al 2019. As a result of this, we now only have one scRNAseq BEC signature because we do not have scRNAseq data from activated human BECS and thus cannot identify a common quiescent and activation BEC signature for both mouse and human.
Our novel scRNAseq signature contains 9 genes (SOX9, MAGI1, BICC1, EPCAM, SLC5A1, DCDC2, ITGB8, KRT7, CYR61) expressed in only human healthy BECs (see new Figure 7A). Next, we validated these genes in scRNAseq data from mouse BECs and confirmed that 8 genes (not Itgb8) were highly expressed in all mouse BECs isolated from healthy or DDC-injured livers (see new Figure 7B). We incorporated these findings into the paper and discared the activated sc LPC and quiescent sc LPC signature which was based on the mouse data from Pepe-Mooney et al. We thank the reviewer for the comment, because we think that this signature is more relevant than our proposed quiescent and activated signature since the new signature is valid in both mouse and human data sets.
We did investigate the data of Segal et al. 2019 and Su et al. 2017. This data contains transcriptomes of foetal human or mouse livers which encompasses a complete different aspect of liver stem cell biology. The focus of our manuscript is on adult liver progenitor cell biology, including developmental data would make the manuscript very complex.
The data of Planas et al., 2019 is indeed very interesting but we could only find the raw data which could not be properly analyzed within the time of the revision. However, we believe that the data of Pepe-Mooney et al., 2019 provides enough validation for the mouse BECs.
Data of MacParland et al., 2018 is generally focused on immune cells and we are not completely convinced about their BEC/Cholangiocyte fraction. When we re-analyze their data there is a low enrichment of BEC genes such as K19 and TROP2 (see attached link with our analyzed data: https://dshare.vub.be:5001/sharing/boB3unWNx ). For these reasons we decided not include this data in our paper although these BECs are also slightly enriched in genes from our scRNAseq signature (see link data).
The lack of validation of new markers assigned to either activated or quiescent LPCs significantly weakens the impact of this study. While it is understood this study is a meta-analysis I would expect to see the validation of novel identified markers – ideally of the scLPC signature of which only 7 genes were identified. The authors should look for expression, either by IHC, IF or in situ RNA hybridization of newly identified activation markers Slco3a1, Wfdc2 and S100a6 in healthy and DDC-injured liver tissue to validate they are specific to activated LPCs.
We understand the remark of the reviewer. However, this would require a major investment in time and money to purchase and test suitable antibodies for IHC. We feel that, although these proteins could be novel markers of BECs and have a functional role in BEC activation, these kind of studies require more time and should be addressed in a separate manuscript.
The authors should also validate the expression of scLPC markers in human cases of AH, AC, NASH and PBC. This would strongly strengthen their claim of distinguishing between activated and quiescent LPCs and increase the confidence in their methodology to identify unique gene signatures. The authors current validation is not sufficient. In supplementary Figure 2 , the authors extract protein atlas staining of novel LPC markers. They do not address that while CDH6, MYO6 and AKAP7 are considered ‘activated markers’ they are enriched in the BECs of non-injured human liver.
By reanalyzing our BEC scRNAseq signature, we now have only one signature for both quiescent and activated BECs for mouse and human. We analyzed the genes from our new scBEC signature in AH, AC, NASH and PSC and provided them as Supplementary figure 4.
We also addressed these results in the discussion as follows:
“Gene signatures can be used to identify certain isolated cell populations. We created a quiescent and activated BEC gene signature that can recognize BEC-like cells and detect BECs in liver diseases accompanied with an activation of the BEC compartment. Strangely, we found no enrichment in PBC which is known to contain expanded BECs [23]. We also investigated whether genes of the scBEC signature could be used to look for BEC activation in the different liver diseases. Only EPCAM, SOX9 and DCDC2 were enriched in ASH and Cirrhotic livers (Supplemental figure 4). This indicates that the scBEC signature is not able to identify activated BECs in bulk transcriptomic data of diseased livers.”
In this study the authors use both human and mouse LPC gene profiles analyzed together to formulate a unique signature. This appears counterproductive to the direction of the field. It is now becoming clear that significant differences appear between mouse and human LPC biology. TROP2 for example is highly expressed in activated LPCS in mouse models of liver disease, but highly expressed in the cholangiocytes of non-injured human liver. While it is fine to use mouse LPC data as a framework for identifying unique LPC markers, the authors should include a figure with more in depth comparison of mouse and human signatures to identify key differences and similarities. It will be interesting to observe if similar quiescent/activation signatures will be present when mouse and human transcriptomic data is separated
Thank you for this comment. We also included gene signatures only expressed by human BECs and quiescent or activated mouse BECs. These gene signatures have been added in supplementary table 2 and are also provided as an XLS file. In addition, we performed gene ontology and Kegg pathway analysis but could not find differences between human and mouse BECs signatures. A more intensive analysis was already performed in figure 3 were we indeed see differences between human and mouse BECs.
The supplementary table 2 was added in the discussion as follows:
“The BEC transcriptomic data obtained from human and mouse livers clearly shows differences. Supplemental table 2 shows a list of genes only expressed in human or mouse BECs which suggest that there are some differences between human and mouse BECs.”
The authors need to clarify in more detail their labelling of ‘quiescent’ and ‘activated’ LPC. It appears to be based on whether the sample is obtained from healthy or injured liver although this is not clearly stated. In Table 1 there are 3 EpCAM+data sets, 1 mouse and 2 human. 1 EpCAM+human data set contains LPCs from injured liver, but this sample is not present within the tsne plot from Figure 1B (or at least not visible). The three TruSeqLT human data sets from Ceulemans, Verhulst et al 2017 do not have accompanying non-injured healthy liver samples (quiescent). As it is likely EpCAM+ and TROP2+FACS sorting will be obtaining BECs/non activated LPCS, it appears they are assuming they are capturing a mostly activated LPC population. The authors then introduce another category for their pathway analysis, ‘initiation’ (by injury inducing diet length). The authors need to make it much clearer which samples specifically refer to a quiescent or activated state (i.e. healthy or injured) and qualify their reasoning and the limitations for this labelling strategy.
We agree with the reviewer that this was not very clear. We now added a new table (supplementary table 1) that identifies which sample was refered to quiescent or activated BECs. We hope that this would clarify which transcriptomic data was used for quiescent or activated BECs.
This was added in material and method as follows:
“For human BEC RNA sequencing data, normalized count files were used from Ceulemans et al. (Table 1)[25]. BEC scRNA seq data was used from Pepe-Mooney et al., 2019 (GSE125688) and Azarina et al., 2019 (GSE124395) [28, 29]. Supplementary table 1 shows a list of sample that refers to quiescent or activated BECs.”
Reviewer 3 Report
The manuscript by Verhulst et al makes novel discoveries regarding unique gene signatures and signaling pathways for quiescent and activated liver progenitor cells (LPCs). Though there is accumulating the evidence regarding the gene signatures of LPCs derived from healthy and injured livers, it was unclear whether these pointed to the same population. Meta-analysis of human and mouse LPC gene profile using available transcriptome data sets revealed that most of LPC populations still possessed highly enriched LPC markers even if different platforms and the isolation techniques have a big impact on LPCs profile. Authors show that KRAS signaling pathway is enriched in LPCs, and it could be important in self-renew and differentiation of LPCs. The quiescent and activated LPC gene signatures that can recognize LPC-like cells and detect LPC in several liver diseases were identified. They also show gene sets that can distinguish quiescent from activated LPCs from a scRNAseq data.
The studies are elegant, carefully designed with use of particular both mouse and human transcriptome data sets.
Minor comment
Authors should cite the manuscript by Schmelzer and Reid et al (“Human stem cells from fetal and postnatal donors” J Exp Med. 2007; 204, 1973-87) as it is representative first evidence regarding characteristic of EPCAM+ human LPCs, even if the cells were not sorted by FACS as single cell level.
Author Response
We thank the reviewer for his enthusiasm for the paper.
Minor comment
Authors should cite the manuscript by Schmelzer and Reid et al (“Human stem cells from fetal and postnatal donors” J Exp Med. 2007; 204, 1973-87) as it is representative first evidence regarding characteristic of EPCAM+ human LPCs, even if the cells were not sorted by FACS as single cell level.
We agree with the reviewer. We’ve added this citation in the introduction (second sentence).
“Early studies in human showed that in conditions of submassive necrosis, reactive ductules and intermediate hepatocyte-like cells originate from the activation and differentiation of putative progenitor cells [1, 2].”